# Effects of Feeding Corn Distillers Dried Grains with Solubles on Muscle Quality Traits and Lipidomics Profiling of Finishing Pigs

**DOI:** 10.3390/ani13243848

**Published:** 2023-12-14

**Authors:** Zhizhuo Ma, Chunsheng Wang, Bo Wang, Linfang Yao, Baohua Kong, Anshan Shan, Jianping Li, Qingwei Meng

**Affiliations:** 1College of Animal Science and Technology, Northeast Agricultural University, Harbin 150030, China; zhizhuoma@163.com (Z.M.); 15765129290@163.com (C.W.); wangbo199901262021@163.com (B.W.); kcylf@163.com (L.Y.); asshan@neau.edu.cn (A.S.); 2College of Food Science, Northeast Agricultural University, Harbin 150030, China; kongbh63@hotmail.com

**Keywords:** finishing pig, distillers dried grains with solubles, meat quality, fatty acid composition, lipidomics

## Abstract

**Simple Summary:**

Corn distillers dried grains with solubles (DDGS) is one of the commonly used feed ingredients in pig production and is usually characterized by high crude fat and high unsaturated fatty acids. However, the effect of adding high levels of DDGS to the diet on the lipid profile of muscle in finishing pigs is unknown. This study aimed to investigate the effects of feeding DDGS on the meat quality, chemical composition, fatty acid composition, and lipidomics profiling in the *longissimus thoracis* (LT) of finishing pigs. The results show that feeding DDGS affects meat quality and fatty acid composition, and alters lipid deposition in muscle by affecting lipid metabolism in finishing pigs.

**Abstract:**

This study investigated the effects of adding corn distillers dried grains with solubles (DDGS) to the diet on the meat quality, chemical composition, fatty acid composition, and lipidomics profiling in the *longissimus thoracis* (LT) of finishing pigs. Twenty-four healthy crossbred pigs (average body weight 61.23 ± 3.25 kg) were randomly divided into two groups with three replicates per group and four pigs per pen. The control group (CON) was fed a basal diet, and the DDGS group was fed an experimental diet with 30% DDGS. The results show that adding DDGS to the diet increases the yellowness (*b**), chroma (*C**), linoleic acid (C18:2n-6) percentages, polyunsaturated fatty acid (PUFA) percentages and iodine value of LT (*p* < 0.05). Based on LC–ESI–MS/MS, 1456 lipids from 6 classes or 44 subclasses in LT were analyzed, and 50 differential lipids were observed. Triglyceride (TG) with C18:2n-6 side chains and ceramide alpha-hydroxy fatty acid-sphingosine (Cer–AS) contents increased significantly, and the decrease in multiple glycerophospholipids (GPs) content may be related to differences in the glycerophospholipid metabolic pathway. Correlation analysis suggests that triglycerides with C18:2n-6 side chains may be one of the reasons for the changes in *b** and *C** values in the LT. In conclusion, feeding DDGS affects the meat quality and fatty acid composition and may affect the lipid profile in the LT of finishing pigs by regulating lipid metabolism.

## 1. Introduction

Growing consumer demand for pork has drawn attention to the shortage of feed resources [1]. Therefore, researchers have begun to study how to use by-products to replace some of the raw materials in the feed without compromising the growth performance and meat quality of pigs. Corn distillers dried grains with solubles (DDGS) is a by-product of ethanol production. During alcohol production, the starch in the grain is converted to ethanol and carbon dioxide, which results in the remaining nutrients such as proteins, fats, fibers, and minerals in DDGS being concentrated nearly three times more compared to the original grain [2]. Because of its rich protein content and low price, corn distillers dried grains with solubles is a suitable substitute for soybean meal in pig diets [3]. Schwarz et al. [4] observed that adding 20% corn distillers dried grains with solubles to the diet did not affect the growth performance and meat quality of pigs. However, corn distillers dried grains with solubles includes a large amount of unsaturated fatty acids, particularly linoleic acid (C18:2n-6), besides being a high-quality protein diet [5]. Previous studies have shown that feeding corn distillers dried grains with solubles can increase C18:2n-6 and polyunsaturated fatty acid (PUFA) in pig muscle [6], and reduce the shelf life and fat hardness of pigs [7]. Another study demonstrated that as the corn distillers dried grains with solubles content in the diet increased, the bellies of pigs became softer and the iodine value in the adipose tissue increased linearly [8]. Most studies have concentrated on how corn distillers dried grains with solubles affected the fatty acid profile in the muscle or different adipose tissues [7,9], meat quality [6], and carcass quality [10] of pigs. However, the specific impact of feeding corn distillers dried grains with solubles on the types of lipids found in meat is still not clearly understood. Therefore, a complete lipid characterization of pork after feeding corn distillers dried grains with solubles is needed to identify altered lipids.

Recently, lipidomics has been widely used in animal husbandry [11]. An increasing number of researchers have started to use lipidomics to identify the lipid composition of pork under different feeding strategies, such as the addition of vegetable oils [12], flaxseed [13], or microalgae [14] to diets, which are of great significance to identify the nutritional value of pork and investigate the effects of different lipids on metabolic regulation of the organism. Due to the influence of corn distillers dried grains with solubles on fatty acid composition and other aspects, we assumed that adding corn distillers dried grains with solubles to the diet could affect the composition and distribution of lipids in pork. Thus, the major aims of this study consisted of analyzing the chemical composition, meat quality, and fatty acid composition of the *longissimus thoracis* (LT) of finishing pigs fed with corn distillers dried grains with solubles, and using a widely targeted lipidomics through a UPLC–ESI–MS/MS system to comprehensively identify lipid molecules to evaluate the changes in pork lipid composition after feeding corn distillers dried grains with solubles.

## 2. Materials and Methods

### 2.1. Animal, Diets, and Treatments

The experimental procedures were approved by the Northeast Agricultural University Laboratory Animal Ethics Committee (NEAUEC20190241) according to the standards described in the “Laboratory Animal Management Regulations” of Heilongjiang Province, China (protocol code 08, 17 October 2008). A total of twenty-four healthy crossbred pigs ([Landrace × Yorkshire] × Duroc, average body weight 61.23 ± 3.25 kg) were randomly allocated into two groups with three replicates per group, and four pigs per pen. The groups were randomized and allocated to one of two dietary treatments, including a control diet with corn and soybean meal as the basis (CON) and an experimental diet with 30% DDGS replacing all soybean meal and some corn and wheat bran in the control diet (DDGS). The composition and nutritional content of the experimental diets are presented in Table 1. The DDGS used for the experiment was full-fat corn DDGS (crude fat: 12.12%; crude protein: 26.49%; crude fiber: 7.61%, Heilongjiang Hongzhan Biotechnology Co., LTD, Qiqihar, China). The fatty acid composition of DDGS is shown in Appendix A. The diets were formulated to meet the nutritional recommendations of the National Research Council (NRC 2012) [15]. During the 55-day experiment, pigs had free availability of diet and water, and the conditions of temperature and humidity were ensured to be suitable. All of the pigs were weighed at the completion of the experiment after a 12 h fast and there were no significant differences between the CON and DDGS groups in the final weight (CON: 108.63 kg vs. DDGS: 107.00 kg, SEM: 1.30 kg) and backfat thickness (CON: 28.07 mm vs. DDGS: 27.06 mm, SEM: 0.85 mm). From each pen, two pigs were chosen at random and sent to the abattoir for slaughter.

### 2.2. Meat Quality Traits

All pigs were processed using slaughterhouse procedures in accordance with the guidelines in Chinese Operating Procedures of Livestock and Poultry Slaughtering—Pig (GB/T 17236-2019) [16]. Samples of LT at the 12th rib of the left side carcass were collected (*n* = 6). A section of each sample was trimmed, vacuum-packaged, and preserved in a 4 °C freezer for meat analysis; another section was snap-frozen in liquid nitrogen and preserved in a −80 °C freezer for lipidomics analysis. The pH value of LT was measured 45 min and 24 h after slaughter using a portable pH meter with automatic temperature compensation (Model 205, Testo AG, Lenzkirch, Germany). The pH meter was calibrated with pH 4.01 and pH 7.00 buffer solutions before the probe was inserted into different parts of the sample for analysis. The temperature compensation function is mainly used to correct the deviation caused by the difference between the temperature at which the standard buffer was used to calibrate the PH meter and the actual sample temperature. The color measurements of lightness (*L**), redness (*a**), and yellowness (*b**) values of the LT after 15 min of blooming at room temperature were measured using a Minolta colorimeter (Minolta Chroma Meter, CR-400; Konica Minolta, Tokyo, Japan) equipped with an 8 mm diameter aperture, a 2° standard observer angle, and a D65 illuminant. Chroma (*C**) was calculated using *a** and *b** measures with the equation: (*a**^2^ + *b**^2^)^1/2^. Hue angle (*h**) was calculated by (arctangent [*b**/*a**]). The bloom means that the vacuum-packed meat was fully in contact with oxygen [17]. The colorimeter was calibrated using a standard white plate before measurement. Three different spots were selected for measurement and the mean value was taken as the result. The method used for drip loss was referenced from Meng et al. [18]. Briefly, duplicate LT samples were cut into 3 cm × 3 cm × 3 cm sizes, and initial muscle weights were recorded. After hanging the sample vertically in a 4 °C freezer for 24 h, surface water was removed and then the final weight was determined. The result was computed by 100× (initial muscle weight − final muscle weight)/initial weight. Cooking loss was obtained from the calculation of the percentage weight loss during the process. Before cooking, surface moisture was wiped off the LT samples, which were then weighed. The LT samples inside bags were cooked in a water bath with 75 °C water until the internal temperature of the samples reached 70 °C. Finally, the samples were cooled at room temperature and weighed. After measuring the cooking loss, the cooked samples were trimmed parallel to the muscle fibers (1 cm × 1 cm × 3 cm). Shear force was then measured using a digital display shear tester (C-LM3, Tenovo, Beijing, China). The device is fitted with a 15 kg load sensor and the crosshead speed is set at 200 mm/min. The shearing action is similar to WarnerBratzler shearing device. By comparing the LT with the American NPPC (2000) colorimetric scoring board, the distribution of intramuscular fat was determined, and the intramuscular fat content increased gradually from 1 to 5 points.

### 2.3. Chemical Composition

The crude fat (960.39), crude protein (981.10), moisture (950.46), and total ash (920.153) contents in LT were analyzed by the AOAC method (AOAC, 1990) [19]. The content of intramuscular fat was analyzed by the Soxhlet extraction method and the intramuscular fat content was presented in percentage of wet muscle weight. After digestion, the crude protein content was computed by the Kjeldahl method. The moisture content was analyzed after drying in an oven. The temperature was 103 °C ± 2 °C, and the result was expressed by percentage. The total ash was burned to gray in a muffle furnace at 550 °C ± 15 °C, cooled to room temperature in a dryer, and then weighed.

### 2.4. Fatty Acid Composition

Referring to the method of Folch, lipids in the LT were extracted with chloroform:methanol (2:1, *v*/*v*) [20]. The extracts were methylated using 0.5 mol sodium methoxy/L in methanol. Fatty acid methyl esters were extracted by hexane, filtered with anhydrous sodium sulfate [21], and then determined using meteorological chromatography (Shimadzu Co., GC-2010, Kyoto, Japan) with a capillary column of SP^TM^-2560 (Supelco Inc., Bellefonte, PA, USA, 100 m × 0.25 mm × 0.20 μm). The gas chromatograph oven temperature was held at 170 °C for 30 min. Then, the temperature was increased at a rate of 1.5 °C/min to a temperature of 200 °C, 5 °C/min to a temperature of 210 °C, and 15 °C/min to a final temperature of 250 °C for 1 min. The injector and flame-ionization detector temperatures were both set at 250 °C. Each fatty acid methyl ester (1 μL) was injected into the split injection port (30:1 split ratio). The fatty acid methyl esters were identified by the comparison of their retention times with an authentic standard. The ionization potential of the mass selective detector was 70 eV, and the scan range was 50 to 550 *m*/*z*. The results were indicated by the area of fatty acid methyl esters that accounted for the total area of all fatty acid methyl esters. The iodine value was calculated as follows: iodine value = [C16:1] × 0.95 + [C18:1] × 0.86 + [C18:2] × 1.732 + [C18:3] × 2.616 + [C20:1] × 0.785 + [C22:1] × 0.723 [22].

### 2.5. Lipidomics Analysis

A 20 mg thawed LT sample was homogenized and 1 mL of extraction solvent (MTBE: MeOH = 3:1, *v*/*v*) containing the internal standard mixture was added. After vortexing and centrifugation, 200 μL of supernatant was evaporated to dryness through a vacuum concentrator (CentriVap, Labconco, Kansas City, MO, USA). Then, the extracts were solved with mobile phase B and detected by LC–MS/MS.

Lipidomics analysis of LT was carried out through the LC–ESI–MS/MS system (UPLC, ExionLC AD, https://sciex.com.cn/, accessed on 8 September 2022; MS, QTRAP^®^ system, https://sciex.com/, accessed on 8 September 2022). The chromatographic column was a Thermo Accucore™ C30 (2.6 μm, 2.1 mm × 100 mm i.d). The mobile phases consisted of A: acetonitrile/water (60/40, *v*/*v*), and B: acetonitrile/isopropanol (10/90 *v*/*v*), and both mobile phases contained 0.1% formic acid and 10 mmol/L ammonium formate. The gradient program for the mobile phases was: A/B (80:20, *v*/*v*) at 0 min, 70:30 *v*/*v* at 2 min, 40:60 *v*/*v* at 4 min, 15:85 *v*/*v* at 9 min, 10:90 *v*/*v* at 14 min, 5:95 *v*/*v* at 15.5 min, 5:95 *v*/*v* at 17.3 min, and 80:20 *v*/*v* at 17.3 min. The flow rate, temperature, and injection volume were 0.35 mL/min, 45 °C, and 2 μL, respectively. Linear ion trap (LIT) and triple quadrupole (QQQ) scans were performed on a QQQ linear ion trap mass spectrometer (QTRAP). The QTRAP^®^ LC–MS/MS system was operated via Analyst 1.6.3 software (AB SCIEX, Toronto, ON, Canada) and had an ESI Turbo Ion-spray interface that can operate in both positive and negative ion modes. The ESI source operating parameters mainly included: ion source temperature of 500 °C, ion source gas 1 of 45 psi, ion source gas 2 of 55 psi, curtain gas of 35 psi, the positive and negative ion mass spectrometry voltages were 5500 V and −4500 V, respectively, and the collision-activated dissociation was medium. Multiple reaction monitoring (MRM) experiments were used to perform QQQ scans with a collision gas of nitrogen at a pressure of 5 psi. The multiple reaction monitoring (MRM) technique, as an analytical method for mass spectrometry detection, enables the targeted selection of data for mass spectrometry signal acquisition while removing interferences from non-compliant ion signals. The declustering potential (DP) and collision energy (CE) for individual MRM transitions were performed with further DP and CE optimization. A specific MRM transition was detected for each duration based on the eluted metabolites.

### 2.6. Data Statistics

SPSS (SPSS standard version 25.0, SPSS Inc., Chicago, IL, USA) software was used to perform independent-sample *t* tests on the data, presented as mean ± standard error of the mean (SEM), and *p* < 0.05 was set to be significant. The mass spectrum (MS) data were analyzed with Analyst 1.6.3 (AB SCIEX, Toronto, ON, Canada). Multivariate and univariate statistical analyses were used to screen differential lipids. The variable importance projection (VIP ≥ 1) and fold change (FC ≥ 2 or ≤ 0.5) of the orthogonal partial least square discriminant analysis (OPLS-DA) model were used as screening conditions.

## 3. Results and Discussion

### 3.1. Meat Quality Traits

The meat quality traits of finishing pigs are presented in Table 2. Adding DDGS to the diet significantly increased the *b** and *C** values of LT (*p* < 0.05). However, the pH_45min_, pH_24h_, *L**, *a**, *h**, cooking loss, drip loss, shear force, and marble score were not influenced by adding DDGS to the diets (*p* > 0.05). The physicochemical properties of meat affect the appearance, shelf life, and sensory quality and contribute to the assessment of the overall level of different meats [23]. In this study, we could not observe significant effects of DDGS on pH, drip loss, cooking loss, shear force, and marble score. Similarly, in the study by Wang et al. [9], the effect of feeding DDGS on these meat quality characteristics was not observed. In addition to being an essential indicator of pork quality, the color of the meat directly influences consumer preference [24]. In the present study, we show that DDGS significantly increases the *b** value of LT, which may lead to a poor sensory experience for consumers, resulting in lower sales of this pork. Similar to our study, previous studies have also shown that feeding DDGS increases the *b** value of animal tissue [25,26]. One possible reason for this result is that feeding DDGS alters fat deposition in animal tissues [25]. In addition, the *C** value is a measure of color saturation and is calculated from the *a** and *b** values. Therefore, the increased *b** and *C** values resulting from feeding DDGS may be related to fat deposition in the LT of pigs.

### 3.2. Chemical Composition

Indexes such as moisture, crude fat, crude protein, and crude ash can reflect the conventional nutrients contained in meat [27]. Table 3 indicates that no significant differences are found in the nutrient composition of the LT between the two groups (*p* > 0.05). Similar to our study, Lee et al. [28] showed that the addition of 20% DDGS to the diet did not affect the chemical composition of the LT in finishing pigs. Smiecinska et al. [23] also failed to observe the differences in the above chemical components after adding DDGS to the diet. This indicates that feeding DDGS does not significantly affect the chemical composition of the LT in finishing pigs.

### 3.3. Fatty Acid Composition

The fatty acid composition of the LT of finishing pigs fed two diets is shown in Table 4. Adding DDGS to the diet significantly increased C18:2n-6 and total PUFA percentages and iodine value (*p* < 0.05). The percentages of myristic acid (C14:0), palmitic acid (C16:0), palmitoleic acid (C16:1), stearic acid (C18:0), oleic acid (C18:1n-9), alpha-linolenic acid (C18:3n-3), eicosadienoic acid (C20:2), eicosatrienoic acid (C20:3), docosahexaenoic acid (C22:6n-3), total SFA, total MUFA, and iodine value in the LT were not influenced by adding DDGS to the diet (*p* > 0.05).

Pork is known to contain a variety of fatty acids, and there is a direct relation between the fatty acid composition of pork and diet [29]. Similar to our result, Xu et al. [7] found that C18:2n-6, C20:2, and PUFA percentages and iodine value in the muscle of growing-finishing pigs were significantly increased by adding DDGS to the diet. In addition, by adding different levels of DDGS to the diet, Benz et al. [30] also found that C18:2n-6 and PUFA percentages and iodine value in the adipose tissue of finishing pigs increased with the increase in DDGS levels. Previous studies have shown that pigs can synthesize SFA and MUFA independently, yet lack the desaturase enzyme used to synthesize linoleic acid, which makes them more susceptible to dietary linoleic acid levels [31,32]. In this study, the contents of C18:2n-6 and PUFA in the DDGS diet were higher than those in the CON diet, whereas there was no significant difference in the amount of fat deposited in the LT between the DDGS and CON groups. Therefore, it was speculated that the higher levels of C18:2n-6 and PUFA in the LT fed DDGS were due to differences in the fatty acid composition of the diets. However, pig muscle is high in C18:2n-6, which may increase the likelihood of inflammatory diseases and affect the overall health of pigs [33]. On the other hand, considering the product consumption perspective, higher levels of PUFA in pork can make pork more susceptible to oxidative rancidity and reduce shelf life [9], which may be disadvantageous for the daily storage of pork. In addition, as the most widely consumed meat, pork is usually made unbalanced in its n-6/n-3 PUFA ratio by conventional feeding methods [34], and the unbalanced n-6/n-3 PUFA ratios are considered to be one cause of cardiovascular disease [35]. Therefore, considering the health aspects, the excessive increase in C18:2n-6 percentage resulting from feeding large amounts of DDGS may reduce the nutritional value of pork.

### 3.4. Lipid Profile Analysis of LT

To further study the lipid changes in the LT, we performed a widely targeted lipidomics analysis of LT based on the LC–ESI–MS/MS system. A total of 1456 lipids were detected (Appendix A). As shown in Figure 1, these lipids are classified into six classes, including 83 fatty acyls (FA) (5.7%), 10 sterol lipids (ST) (0.69%), 187 sphingolipids (SP) (12.84%), 2 prenol lipids (PR) (0.14%), 391 glycerolipids (GL) (26.85%), and 783 glycerophospholipids (GP) (53.78%). These lipids were further divided into 44 subclasses, and it was found that the most abundant lipids were triglyceride (TG) and different types of GPs. As an important source of fatty acids and energy, pork contains several classes of lipids [36,37]. The report of Mi et al. [36] showed that GP (452, 38.31%) and GL (242, 20.51%) were the two lipids with the largest number of pork lipids identified by the UPLC–ESI–MS/MS system. In addition, Li et al. [38] indicated that the metabolism of GP and GL is the keyway to regulating IMF, which is related to meat quality. Therefore, GP and GL may be the main lipids regulating fat deposition. By further analyzing the lipid composition, we observed that TG accounted for 22.05% of all lipid subclasses and was the most abundant lipid in the LT fed DDGS. It has been reported that TG is mainly stored in lipid droplets in muscle [39], and GP is thought to be mainly from the cell membrane [40,41]. Intramuscular triglycerides are energy supply and dynamic fat-storage depots and play a crucial role in the metabolism of the body [42]. In addition, increased triglycerides in muscle have also been suggested to be associated with obesity in pigs [43].

### 3.5. Differential Lipid Analysis

To further expand the differences between the groups, it was beneficial to find the differential lipids between the CON and DDGS groups. We established an OPLS-DA model to analyze the LT samples further. As shown in Figure 2A, the OPLS-DA score plot shows a significant distinction between the two groups, which can better distinguish the samples under two different feeding strategies. Fitness (R^2^Y = 0.964) and predictability (Q^2^ = 0.564) show that the model is effective and reliable (Appendix A). According to the OPLS-DA findings, a volcano map was drawn to show the differences of lipids in LT more visually and clearly under different feeding strategies. A total of 50 lipids that were significantly different between the two groups were identified by combining the variable importance projection (VIP) values and the fold change (FC) values (VIP ≥ 1; FC ≥ 2 or FC ≤ 0.5) (Figure 2B). The 50 differential lipids are listed in Table 5. A total of 14 of these lipids were up-regulated (including ceramide alpha-hydroxy fatty acid-sphingosine (Cer–AS), phosphatidic acid (PA), phosphatidylinositol (PI), and TG) and 36 lipids were down-regulated (including acylcarnitine (CAR), bile acid (BA), diacylglycerophosphoethanolamines (LNAPE), lysophophatidylcholine (LPC), lysophosphatidylglycerol (LPG), phosphatidic acid (PA), phosphatidylcholine (PC), phosphatidylethanolamine (PE), alkylglycerophosphoethanolamines (PE-O), alkenylglycerophosphoethanolamines (PE-P), phosphatidylglycerol (PG), phosphatidylinositol (PI), phosphatidylmethanol (PMeOH), phosphatidylserine (PS), and sphingomyelin (SM)) in the DDGS group. Figure 2C–F show the contents of representative differential lipids among them. In addition, the clustering heat map assists in observing the variation in lipid content in each LT sample (Figure 3). The figure shows that the lipid molecules with the highest increase in the content are mainly Cer–AS and TG, whereas the contents of various types of GPs are significantly down-regulated.

As a bioactive membrane sphingolipid, ceramide (Cer) plays an important role in regulating different metabolic pathways according to its subcellular distribution and acyl chain length [44]. In the presence of excess lipids in the organism, Cer promotes the conversion of excess free fatty acids (FFA) into TG for storage and accelerates the utilization of FFA [45]. It was reported that a high-fat diet can significantly increase the content of Cer in the muscle of mice [46]. In this study, the crude fat content of the DDGS diet was higher than the CON diet, which may have contributed to the increased Cer content in the LT of finishing pigs. Meanwhile, we found that the high-fat diet enriched in C18:2n-6 was more likely to induce an increase in the content of the Cer–AS subclass compared to other types of Cer. Cer–AS has been demonstrated to be one of the constituents of the human and pig stratum corneum and plays an important role in maintaining the skin barrier as well as protecting pigs from toxic substances, allergens, and pathogenic microorganisms in the environment [47,48]. Therefore, adding DDGS to the diet may promote the growth of the cuticles, thereby benefiting the growth of finishing pigs under poorer feeding conditions. In addition, as a common feed for pigs, DDGS is rich in PUFA, especially C18:2n-6 [5]. In this study, the contents of most TG with C18:2n-6 side chains (mainly including TG(10:0_16:2_18:2), TG(14:0_18:2_18:2), TG(16:0_18:2_20:4), TG(17:1_18:2_18:3), TG(18:2_18:2_18:3), and TG(9:0_18:1_18:2)) in LT of finishing pigs were increased by feeding DDGS, which was consistent with the results detected by gas chromatography. A previous study shows that dietary linoleic acid becomes more incorporated into the side chains of triglycerides and less into phospholipids [49], which is similar to the present results. Another study showed that dietary intervention had a more marked effect on phospholipids in pig muscle than TG [12]. In the present study, we observed that feeding DDGS led to an increase in the content of TG with C18:2n-6 side chain in the LT of finishing pigs. One possible explanation for this finding is that although GP is more susceptible to dietary PUFA, excess C18:2n-6 is mainly stored in muscle TG due to the limited ability of GP to bind C18:2n-6 [49]. Furthermore, similar to the previous study [50], we did not observe significant changes in backfat thickness in finishing pigs fed DDGS. This may suggest that neutral lipid fatty acids and phospholipids are not differentially deposited in different tissues of finishing pigs. On the other hand, C18:2n-6 is essential for human health as an essential fatty acid that influences cell membrane fluidity and the behavior of membrane-binding enzymes and receptors [51]. However, nutritional research has found that excessive consumption of foods containing C18:2n-6 may lead to the development of metabolic diseases and inflammation [52]. Notably, although dietary TG is essential for human energy metabolism and systemic lipid homeostasis, excessive intake of TG can cause lipid metabolism disorders and increase the risk of obesity [53]. Therefore, feeding DDGS leads to an increase in n-6 long-chain triglycerides in pork, and excessive consumption of this pork may have potentially negative effects on human health.

In addition, the contents of various types of GPs in LT were significantly down-regulated by feeding DDGS. A recent study indicated that when n-3 PUFA-rich microalgae were added to pig diets, the content of GP with double bond ≤ 4 decreased, while the content of GP containing docosahexaenoic acid (C22:6) and eicosapentaenoic acid (C20:5) acyl chain increased significantly [14]. Similar to this study, we observed that the addition of DDGS to the diet decreased the content of most of the GPs containing non-C18:2n-6 in LT, which suggests that DDGS affects the fatty acid composition of LT mainly by influencing the lipoyl side chains of TG and GP. Interestingly, other feed ingredients such as vegetable oils have, likewise, been shown to influence the fatty acid composition of muscle by affecting the lipoyl side chains of phospholipids and triglycerides in the *longissimus* muscle of finishing pigs [12], which is similar to the effects we observed for DDGS.

Since the addition of 30% DDGS to the diet alters the meat color, fatty acid composition, and lipid composition of LT, we suggest that the addition of DDGS can be reduced in the diet to mitigate these changes. In addition, previous studies have shown that the *b** value and linoleic acid levels can be reduced by means of nutritional modulation, such as the addition of conjugated linoleic acid [6,54] or betaine [6,55] to DDGS. Therefore, these methods may be beneficial in alleviating the negative effects of adding high levels of DDGS to the diet.

### 3.6. Pathway Analysis

As shown in Figure 4A, in the organismal systems network, 27 differential lipids (84.38%) were enriched in metabolic pathways, 22 differential lipids (68.75%) were enriched in glycerophospholipid metabolism, and 14 differential lipids (43.75%) were enriched in retrograde endocannabinoid signaling. In addition, in the enrichment analysis plot (Figure 4B), the top 20 pathways of *p*-value were screened for display. The pathways with higher enrichment were mainly glycerophospholipid metabolism, retrograde endocannabinoid signaling, and choline metabolism in cancer. By analyzing the metabolic pathways in LT, we speculated that the significant decreases in the content of multiple GPs may be due to the differences in the glycerophospholipid pathway. GP is a crucial lipid in almost all mammalian cell membranes and contains various types of subclasses with functions such as forming cell membranes, regulating transport, and participating in signaling [56]. In addition, in this study, we also found that feeding DDGS significantly affected retrograde endocannabinoid signaling in LT. Similar to our study, Alvheim et al. [57] observed that high dietary content of C18:2n-6 significantly increased endocannabinoid levels in animal livers. Endocannabinoids influence homeostatic, appetite-trigger food intake, and are closely related to dietary levels of C18:2n-6 and arachidonic acids [58]. A diet containing high contents of C18:2n-6 may contribute to obesity by affecting endocannabinoid signaling [57].

### 3.7. Correlation Analysis

The Spearman correlation analysis of the differential lipids and meat quality traits is shown in Figure 5 and Appendix A. Among them, major meat color traits such as *b** and *C** values are positively correlated with TG and Cer–AS, and negatively correlated with various GPs. A previous study shows that increased yellowness of pig adipose tissue is associated with C18:2n-6 [59]. In the present study, C18:2n-6 was predominantly deposited in the TG, leading to an increase in the n-6 long-chain triglyceride content, which may be one of the reasons for the increase in *b** and *C** values of the LT.

## 4. Conclusions

In summary, this study suggests that the addition of DDGS to the diet altered the meat quality, fatty acid composition, and lipid composition of LT.

From a product consumption perspective, this may reduce consumer desire to purchase this pork and increase the potential for oxidative rancidity. In addition, from a consumer health perspective, consumption of large quantities of this pork may have potentially negative effects on health. These results provide a more detailed analysis of the effects of feeding DDGS on lipid molecules in LT of pigs, and provide some insights for analyzing the effects of different pork on consumer health and improving pork quality.

## Figures and Tables

**Figure 1 animals-13-03848-f001:**
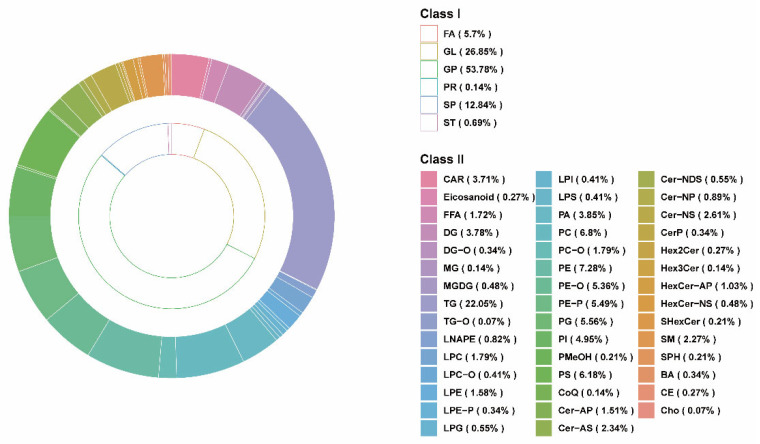
Lipidomics analysis of *longissimus thoracis* samples. The total lipid composition of classes and subclasses in the CON and DDGS groups, with the inner circle representing classes and the outer circle representing subclasses. CON, control group; DDGS, DDGS group.

**Figure 2 animals-13-03848-f002:**
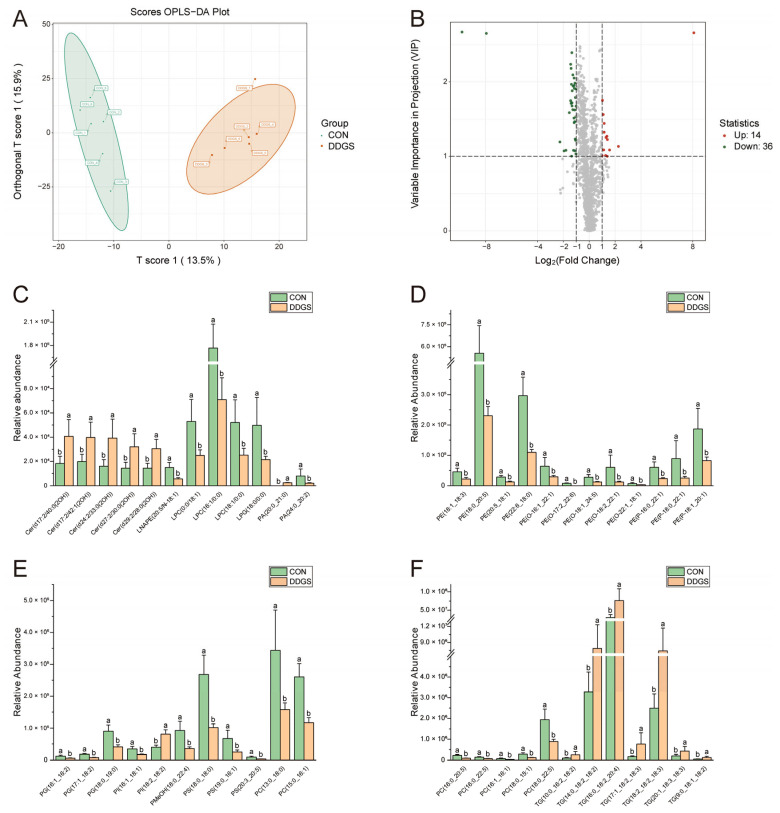
Analysis of lipids with significant differences between CON and DDGS groups. (**A**) Orthogonal partial least squares-discriminant analysis (OPLS-DA) of *longissimus thoracis* lipid profile. (**B**) Differential lipid volcano plot. (**C**–**F**) Representative differential lipids. T score 1 denotes the score of the main components in the orthogonal signal collection process, and orthogonal T score 1 denotes the score of the orthogonal component in the orthogonal signal collection process. The red dots indicate that the content is significantly up-regulated, the green dots indicate that the content is significantly down-regulated, and the gray dots indicate that the content is not significant. CON, control group; DDGS, DDGS group. Data were expressed as mean ± standard error of the mean, *n* = 6. ^a,b^ Different letters indicate significant differences (VIP ≥ 1; FC ≥ 2 or FC ≤ 0.5).

**Figure 3 animals-13-03848-f003:**
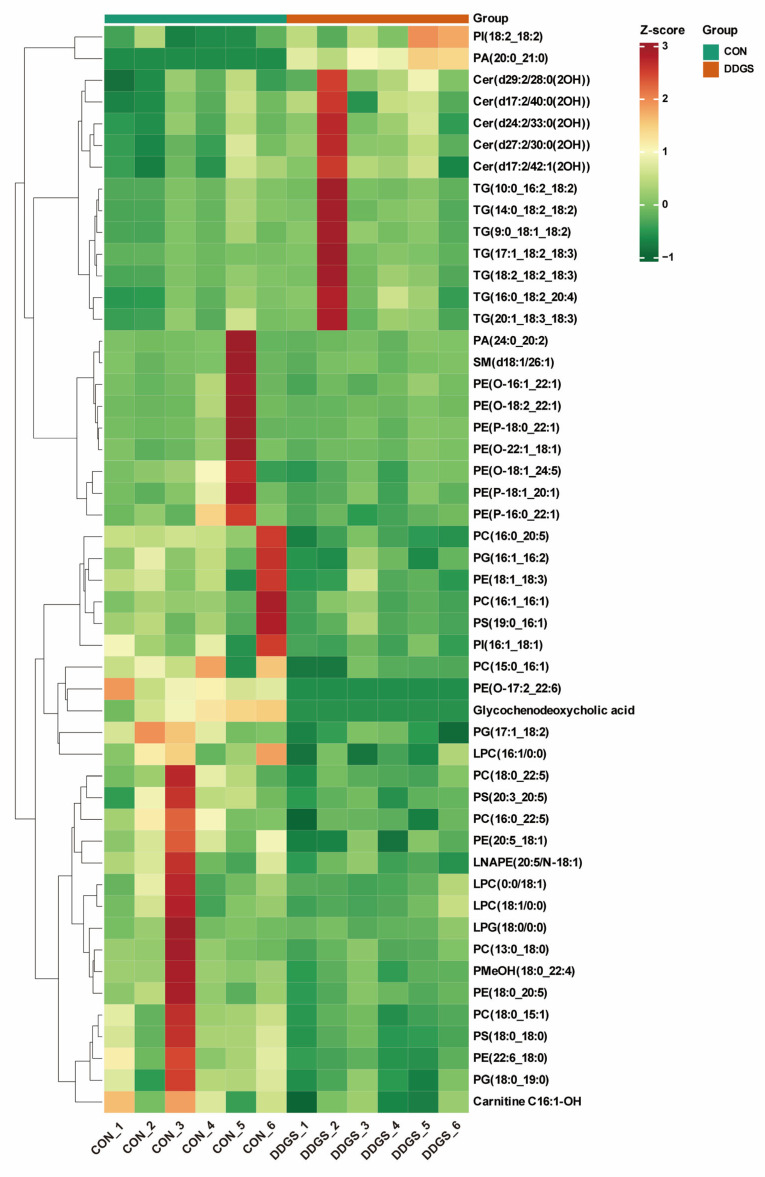
Differential lipid clustering heat map between CON and DDGS groups. Color represents lipid content, with darker red indicating higher lipid content and darker green indicating lower lipid content. Samples clustered in the same cluster by clustering lines have higher similarity between samples. CON, control group; DDGS, DDGS group, *n* = 6.

**Figure 4 animals-13-03848-f004:**
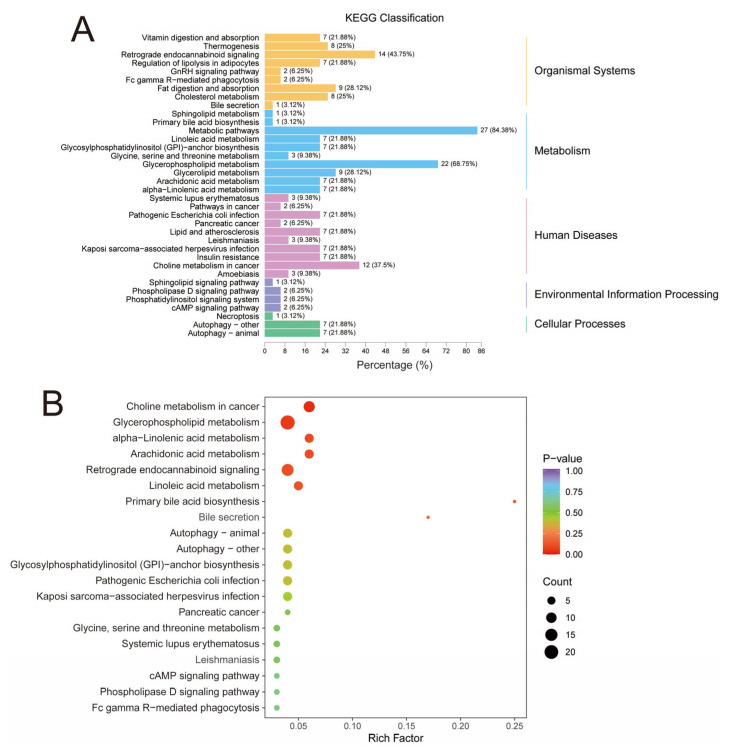
Differential lipid KEGG pathway analysis. (**A**) KEGG classification. (**B**) KEGG enrichment analysis. The enrichment degree was comprehensively expressed by the rich factor, *p*-value, and the number of differential lipids in the corresponding pathway. The rich factor is the ratio of the number of differential lipids in the corresponding pathway to the total number of lipids annotated in the pathway. The greater the value, the higher the degree of enrichment. The closer the *p*-value is to 0 (the redder the color of the point), the higher the enrichment. The size of the point represents the number of differentially enriched lipids. The larger the shape, the more differentially enriched lipids.

**Figure 5 animals-13-03848-f005:**
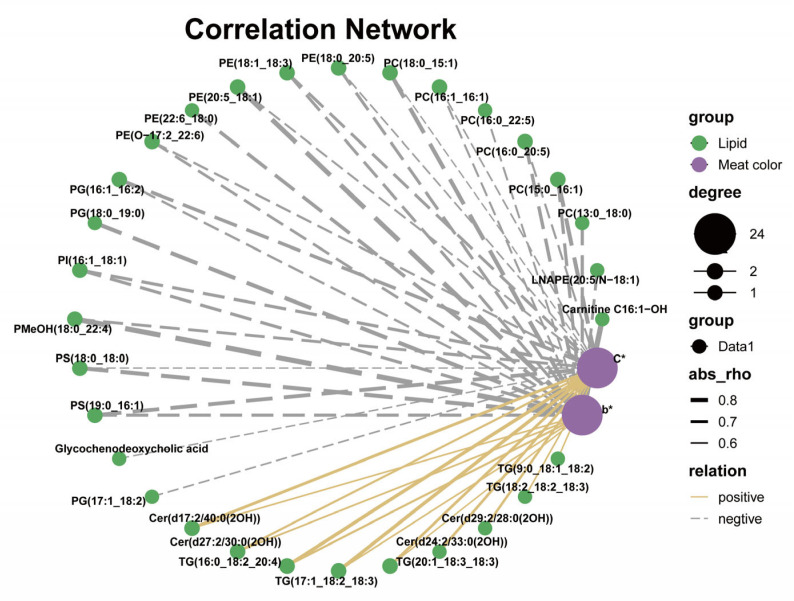
Spearman correlation analysis of differential lipids and meat color in *longissimus thoracis* (*n* = 6). The solid yellow line and gray dotted line represent positive and negative correlations, respectively. In addition, the line width indicates the strength of correlation. Only the significant edges are drawn in the network using the Spearman correlation test (0.5 < |*r*| < 1; *p* < 0.05).

**Table 1 animals-13-03848-t001:** Composition and nutrient levels of basal diets (as-fed basis).

Item	Levels of DDGS Supplementation	
	0%	30%
Ingredients (%)		
Corn	73.69	62.58
Soybean meal	17.00	−
DDGS	−	30.00
Wheat bran	4.18	1.80
Soybean oil	1.40	1.40
Dicalcium phosphate	0.98	0.67
Limestone	0.80	1.12
L-lysine-HCL	0.38	0.77
DL-methionine	0.06	0.03
L-threonine	0.10	0.16
L-tryptophan	0.02	0.08
Choline chloride	0.09	0.09
NaCL	0.30	0.30
Vitamin and mineral premix ^1^	1.00	1.00
Total	100.00	100.00
Nutrient levels ^2^, (%)		
Net energy, MJ/kg	10.36	10.36
Crude protein	14.49	14.44
Crude fat	4.52	6.73
Crude fiber	2.98	3.54
Calcium	0.59	0.59
Total phosphorus	0.52	0.52
Available phosphorus	0.25	0.30
SID lysine	0.85	0.85
SID tryptophan	0.15	0.15
SID methionine	0.27	0.27
SID threonine	0.52	0.52
Fatty acid composition, % of total fatty acid		
Myristic acid C14:0	0.14	0.26
Palmitic acid C16:0	9.38	8.99
Palmitoleic acid C16:1	0.26	0.22
Stearic acid C18:0	3.26	3.03
Oleic acid C18:1n-9	19.71	21.51
Linoleic acid C18:2n-6	47.29	49.88
α-Linolenic acid C18:3n-3	0.50	0.37
Eicosadienoic acid C20:2	0.25	1.16
Eicosatrienoic acid C20:3	0.70	1.50
Docosahexaenoic acid C22:6n-3	0.57	0.67
Total SFA ^3^	15.15	14.14
Total MUFA ^4^	24.28	23.67
Total PUFA ^5^	48.37	51.40

DDGS, corn distillers dried grains with solubles; SID, standardized ileal digestible. ^1^ Vitamins and minerals per kilogram of diet: vitamin A, 8000 IU; vitamin D_3_, 2000 IU; vitamin E, 30 IU; vitamin K_3_, 1.5 mg; vitamin B_1_, 1 mg; vitamin B_2_, 3 mg; vitamin B_6_, 1 mg; vitamin B_12_, 12 µg; folic acid, 0.3 mg; niacin, 15 mg; calcium pantothenate, 12 mg; Fe, 100 mg as FeSO_4_·H_2_O; Cu, 20 mg as CuSO_4_·5H_2_O; Mn, 26.6 mg as MnSO_4_·H_2_O; Zn, 99.4 mg as ZnSO_4_·H_2_O; Se, 0.2 mg as Na_2_SeO_3_; and I, 0.3 mg as KI. ^2^ Nutrition levels were calculated by using the tables of feed composition and nutritive values in China (2020, thirty-first edition) Chinese feed database according to the chemical composition of the dietary ingredients. ^3^ Total SFA = [C8:0] + [C10:0] + [C12:0] + [C14:0] + [C16:0] + [C17:0] + [C18:0] + [C20:0] + [C22:0] + [C24:0]. ^4^ Total MUFA = [C14:1] + [C16:1] + [C18:1n-9] + [C18:1n-7] + [C20:1] + [C24:1]. ^5^ Total PUFA = [C18:2n-6] + [C18:3n-3] + [C18:3n-6] + [C20:2] + [C20:4n-6].

**Table 2 animals-13-03848-t002:** Effect of feeding DDGS on the meat quality of the *longissimus thoracis* of finishing pigs.

Items	CON	DDGS	SEM	*p*-Value
pH_45min_	6.14	5.97	0.10	0.415
pH_24h_	5.72	5.60	0.04	0.117
Lightness (*L**)	45.70	46.00	0.72	0.846
Redness (*a**)	5.05	6.84	0.55	0.103
Yellowness (*b**)	7.09 ^b^	7.81 ^a^	0.18	0.035
Chroma (*C**) ^1^	8.79 ^b^	11.22 ^a^	0.61	0.038
Hue angle (*h**) ^1^	55.27	46.22	2.86	0.117
Cooking loss, %	29.66	33.79	1.81	0.275
Drip loss, %	3.65	4.02	0.35	0.624
Shear force ^2^, N	85.49	85.37	7.02	0.993
Marbling score	2.67	2.83	0.28	0.780

CON, a control diet based on corn and soybean meal; DDGS, corn distillers dried grains with solubles; SEM, standard error of the mean. ^a,b^ Values in the same row with different superscript letters were significantly different (*p* < 0.05), *n* = 6. ^1^ Chroma (*C**) was calculated using *a** and *b** measures with the equation: (*a**^2^ + *b**^2^)^1/2^. Hue angle (*h**) was calculated by (arctangent [*b**/*a**]). ^2^ Shear force was measured using a digital shear apparatus (C-LM3, Tenovo, Beijing, China).

**Table 3 animals-13-03848-t003:** Effect of feeding DDGS on the chemical composition of the *longissimus thoracis* of finishing pigs.

Items	CON	DDGS	SEM	*p*-Value
Moisture (g/100 g meat)	71.66	70.79	0.44	0.349
Fat (g/100 g meat)	2.56	2.47	0.19	0.834
Protein (g/100 g meat)	23.27	23.7	0.42	0.636
Ash (g/100 g meat)	1.22	1.27	0.02	0.258

CON, a control diet based on corn and soybean meal; DDGS, corn distillers dried grains with solubles; SEM, standard error of the mean, *n* = 6.

**Table 4 animals-13-03848-t004:** Effect of feeding DDGS on the fatty acid composition of the *longissimus thoracis* of finishing pigs (g/100 g total fatty acids).

Items	CON	DDGS	SEM	*p*-Value
Myristic acid (C14:0)	1.04	1.07	0.033	0.695
Palmitic acid (C16:0)	22.19	22.31	0.589	0.925
Palmitoleic acid (C16:1)	2.23	2.13	0.153	0.760
Stearic acid (C18:0)	12.83	12.03	0.309	0.216
Oleic acid (C18:1n-9)	41.46	39.64	0.840	0.305
Linoleic acid (C18:2n-6)	7.89 ^b^	11.98 ^a^	1.048	0.042
alpha-Linolenic acid (C18:3n-3)	0.33	0.28	0.037	0.537
Eicosadienoic acid (C20:2)	0.24	0.39	0.071	0.314
Eicosatrienoic acid (C20:3)	0.57	0.51	0.12	0.816
Docosahexaenoic acid (C22:6n-3)	0.40	0.43	0.09	0.855
Total SFA ^1^	38.78	37.68	0.635	0.418
Total MUFA ^2^	44.72	42.81	0.786	0.246
Total PUFA ^3^	8.46 ^b^	12.64 ^a^	1.089	0.046
Iodine value ^4^, g/100 g	16.71 ^b^	23.69 ^a^	1.755	0.037

CON, a control diet based on corn and soybean meal; DDGS, corn distillers dried grains with solubles; SEM, standard error of the mean. ^a,b^ Values in the same row with different superscript letters were significantly different (*p* < 0.05), *n* = 6. ^1^ SFA = saturated fatty acid; total SFA = [C8:0] + [C10:0] + [C12:0] + [C14:0] + [C16:0] + [C17:0] + [C18:0] + [C20:0] + [C22:0] + [C24:0]. ^2^ MUFA = monounsaturated fatty acid; total MUFA = [C14:1] + [C16:1] + [C18:1] + [C18:1n-7] + [C20:1] + [C24:1]. ^3^ PUFA = polyunsaturated fatty acid; total PUFA = [C18:2] + [C18:3] + [C18:3] + [C20:2] + [C20:4]. ^4^ Iodine value = [C16:1] × 0.95 + [C18:1] × 0.86 + [C18:2] × 1.732 + [C18:3] × 2.616 + [C20:1] × 0.785 + [C22:1] × 0.723.

**Table 5 animals-13-03848-t005:** Significantly different lipid molecules between CON and DDGS group.

Compounds	Class I	Class II	VIP	Fold Change	Log_2_FC	Type
Carnitine C16:1-OH	FA	CAR	1.79	0.48	−1.07	Down
Cer(d17:2/40:0(2OH))	SP	Cer–AS	1.32	2.22	1.15	Up
Cer(d17:2/42:1(2OH))	SP	Cer–AS	1.01	2.00	1.00	Up
Cer(d24:2/33:0(2OH))	SP	Cer–AS	1.25	2.45	1.30	Up
Cer(d27:2/30:0(2OH))	SP	Cer–AS	1.44	2.23	1.16	Up
Cer(d29:2/28:0(2OH))	SP	Cer–AS	1.56	2.12	1.09	Up
Glycochenodeoxycholic acid	ST	BA	2.65	0.00	−7.95	Down
LNAPE(20:5/N-18:1)	GP	LNAPE	1.71	0.38	−1.40	Down
LPC(0:0/18:1)	GP	LPC	1.23	0.47	−1.08	Down
LPC(16:1/0:0)	GP	LPC	1.88	0.40	−1.31	Down
LPC(18:1/0:0)	GP	LPC	1.22	0.48	−1.05	Down
LPG(18:0/0:0)	GP	LPG	1.08	0.43	−1.21	Down
PA(20:0_21:0)	GP	PA	2.66	271.17	8.08	Up
PA(24:0_20:2)	GP	PA	1.07	0.26	−1.94	Down
PC(13:0_18:0)	GP	PC	1.47	0.46	−1.12	Down
PC(15:0_16:1)	GP	PC	1.95	0.45	−1.16	Down
PC(16:0_20:5)	GP	PC	2.39	0.39	−1.34	Down
PC(16:0_22:5)	GP	PC	1.91	0.49	−1.02	Down
PC(16:1_16:1)	GP	PC	1.72	0.40	−1.33	Down
PC(18:0_15:1)	GP	PC	2.09	0.42	−1.27	Down
PC(18:0_22:5)	GP	PC	1.62	0.46	−1.13	Down
PE(18:0_20:5)	GP	PE	1.63	0.42	−1.27	Down
PE(18:1_18:3)	GP	PE	1.53	0.49	−1.03	Down
PE(20:5_18:1)	GP	PE	2.05	0.44	−1.17	Down
PE(22:6_18:0)	GP	PE	2.24	0.37	−1.43	Down
PE(O-16:1_22:1)	GP	PE-O	1.07	0.46	−1.11	Down
PE(O-17:2_22:6)	GP	PE-O	2.67	0.00	−9.83	Down
PE(O-18:1_24:5)	GP	PE-O	1.45	0.45	−1.16	Down
PE(O-18:2_22:1)	GP	PE-O	1.19	0.21	−2.26	Down
PE(O-22:1_18:1)	GP	PE-O	1.03	0.49	−1.03	Down
PE(P-16:0_22:1)	GP	PE-P	1.68	0.40	−1.33	Down
PE(P-18:0_22:1)	GP	PE-P	1.08	0.29	−1.80	Down
PE(P-18:1_20:1)	GP	PE-P	1.51	0.44	−1.18	Down
PG(16:1_16:2)	GP	PG	1.98	0.47	−1.10	Down
PG(17:1_18:2)	GP	PG	1.94	0.42	−1.24	Down
PG(18:0_19:0)	GP	PG	1.91	0.45	−1.15	Down
PI(16:1_18:1)	GP	PI	1.68	0.50	−1.01	Down
PI(18:2_18:2)	GP	PI	1.75	2.03	1.02	Up
PMeOH(18:0_22:4)	GP	PMeOH	1.97	0.39	−1.37	Down
PS(18:0_18:0)	GP	PS	2.18	0.38	−1.40	Down
PS(19:0_16:1)	GP	PS	1.75	0.37	−1.43	Down
PS(20:3_20:5)	GP	PS	1.59	0.35	−1.54	Down
SM(d18:1/26:1)	SP	SM	1.00	0.39	−1.36	Down
TG(10:0_16:2_18:2)	GL	TG	1.00	2.63	1.39	Up
TG(14:0_18:2_18:2)	GL	TG	1.01	2.41	1.27	Up
TG(16:0_18:2_20:4)	GL	TG	1.27	2.59	1.37	Up
TG(17:1_18:2_18:3)	GL	TG	1.13	4.78	2.26	Up
TG(18:2_18:2_18:3)	GL	TG	1.09	2.97	1.57	Up
TG(20:1_18:3_18:3)	GL	TG	1.09	2.17	1.12	Up
TG(9:0_18:1_18:2)	GL	TG	1.23	2.65	1.40	Up

CON, a control diet based on corn and soybean meal; DDGS, corn distillers dried grains with solubles; VIP, variable importance in projection; fold change, calculating the fold change by dividing the average relative content of DDGS group by the average relative content of CON group.

## Data Availability

The data presented in this study are available in the present article and are shared with consent and in accordance with all authors.

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
