# Peer review of "Effects of Feeding Corn Distillers Dried Grains with Solubles on Muscle Quality Traits and Lipidomics Profiling of Finishing Pigs"

_animals, 2023, doi:10.3390/ani13243848_

Round 1

Reviewer 1 Report

Comments and Suggestions for Authors

The article is based on an experiment whose number of replicates could be higher. Still, the study is valid, but obviously has less statistical sensitivity.

Some mention should be made of the animal performance, after all it only makes sense to analyze carcass quality if the treatment under test proved to be minimally viable in terms of zootechnical performance.

The test ingredient evaluated, DDGS, should be characterized. Information is needed regarding the basic composition such as protein and crude fat content (and if possible, fatty acid profile). It is worth to mention that there are different types of DDGS, fat could be extracted. This does not seem to be the case with the product tested here, since its inclusion at 30% increased the crude fat content from 3.5 to 6.7%. This higher fat content (compared to corn and soybean meal that gave way to it in the formula) was probably decisive in terms of the effect on the fatty acid profile of the animals' fat. But the result must be interpreted with caution, as depending on the DDGS (especially if degreased), the result can be different.

Overall the article is very well written and the fat profile comparison was explored with advanced techniques. What can still be improved: In the discussion, clarify the real importance of the observed results, that is, what advantage or disadvantage there is in producing pigs whose carcass quality changes in the b value and in the observed fatty acid profile.

Reviewer 2 Report

Comments and Suggestions for Authors

This study explored the effects of DDGS replacing soybean meal and some corn and wheat bran on the quality and composition of meat in finishing pigs, and made a comprehensive and in-depth analysis of lipid metabolism. The findings have reference significance in intramuscular fat regulation. The manuscript is well-written. There are no errors in the figures and tables. However, I think there are still some issues that should be addressed or improved before considering acceptation for publication.

Main comments:

1.     How about the growth performance of pigs after dietary DDGS supplementation?

2.     According to this study, feeding DDGS leads to increased b* value and n-6 long-chain triglycerides in pork. Does this mean that 30% DDGS in feed is too high for pigs? Is there any method to alleviate these negative impacts of DDGS?

3.     “The addition of DDGS to the diet may be beneficial to the growth of cuticles in finishing pigs”. How will the growth of cuticles affect pig production? Is it harmful or beneficial?

4.     “DDGS affects the fatty acid composition of LT mainly by influencing the lipoyl side chains of TG and GP”. Is there any other feed ingredients have similar roles?

Specific comments:

L83-84: Is the nutrient levels calculated by NRC?

L86: What is the “conventional slaughterhouse procedures”? What is the animal ethics approval number of this experiment?

L95: Is this "blooming" misspelled? What does it mean?

L100: reference to…

L111: Please add the manufacturer information of the shear tester.

L159: “Declustering”.

L350: “mainly stored”.

L359: Is this “effects” positive or negative? Please rewrite this sentence.

Comments on the Quality of English Language

This study explored the effects of DDGS replacing soybean meal and some corn and wheat bran on the quality and composition of meat in finishing pigs, and made a comprehensive and in-depth analysis of lipid metabolism. The findings have reference significance in intramuscular fat regulation. The manuscript is well-written. There are no errors in the figures and tables. However, I think there are still some issues that should be addressed or improved before considering acceptation for publication.

Main comments:

1.     How about the growth performance of pigs after dietary DDGS supplementation?

2.     According to this study, feeding DDGS leads to increased b* value and n-6 long-chain triglycerides in pork. Does this mean that 30% DDGS in feed is too high for pigs? Is there any method to alleviate these negative impacts of DDGS?

3.     “The addition of DDGS to the diet may be beneficial to the growth of cuticles in finishing pigs”. How will the growth of cuticles affect pig production? Is it harmful or beneficial?

4.     “DDGS affects the fatty acid composition of LT mainly by influencing the lipoyl side chains of TG and GP”. Is there any other feed ingredients have similar roles?

Specific comments:

L83-84: Is the nutrient levels calculated by NRC?

L86: What is the “conventional slaughterhouse procedures”? What is the animal ethics approval number of this experiment?

L95: Is this "blooming" misspelled? What does it mean?

L100: reference to…

L111: Please add the manufacturer information of the shear tester.

L159: “Declustering”.

L350: “mainly stored”.

L359: Is this “effects” positive or negative? Please rewrite this sentence.

Reviewer 3 Report

Comments and Suggestions for Authors

This study adressed an important point for the guidelines in animal nutrition. DDGS is a by-product used in various animal productions such as poultry. Determining what would be the appropiate proportion in feeding pigs would determine its correct use and nutritional qualities, as well as the resulting benefits on the product obtained.  

The figures used to show the results are very well and interesting. It is suggested to incorporated in the conclusions  a complementary parragraph on the consumption of the product and these effects on the consumer´s health.

Paper review: “Effects of feeding corn distillers dried grains with solubles on muscle quality traits and lipidomics profiling of finishing pigs”

Line 19: it is suggested to write the number twenty-four with digits (24). Same in line 66.

Line 19: replace word “gilts” for “pig” (standardize the nomenclature). Same in line 66.

Line 49: Maybe you could rephrase “feeding DDGS alters explicitly what type of lipids in the meat is still unclear” to “The specific impact of feeding DDGS on the types of lipids found in meat is still not clearly understood.”

Line 61: it is suggested to add the word “system” after UPLC-ESI-MS/MS

Line 86: in the sentence “all pigs were slaughtered following conventional slaughterhouse procedures” consider rewriting to avoid the negative connotations associated with “slaughtered”. For instance, “all pigs were processed using conventional slaughterhouse procedures.”

Introduction

Avoid excessive use of abbreviation such as “DDGS” and “UFA”. Even though they are defined in the text, repeated use might make reading challenging.

Materials and methods

avoid excessive use of technical jargon without explanation. While it is a methodology section and technical language is to be expected, it is helpful for the reader to have a brief explanation or reference for less common methods.

Consider adding more detail or references for methods or procedures that might not be familiar to all readers.

It is good that detailed information on the experimental diets and their composition has been provided.

The materials and methods section appears to be well written.

Conclusion

In conclusion, the study has provided valuable insights into the impact of DDGS supplementation led to significant increases in certain fatty acids and triglycerides, it also resulted in notable reductions in various glycerophospholipids. Given the observed alterations in lipid profiles and meat quality when feeding DDGS to finishing pigs, as highlighted in this study, could you elucidate on the implications of these changes? Specifically, are these changes beneficial or detrimental to the health and well-being of the pigs? Additionally, how might these modifications impact the nutritional quality and safety for consumers?

The potential implications of these changes on meat quality, nutritional value, and overall health of the animals warrant further investigation. Moreover, considering the potential health implications for human consumers, a thorough understanding of these changes is essential to guide future dietary formulations and recommendations.

Reviewer 4 Report

Comments and Suggestions for Authors

Peer Review report 1 on: “Effects of feeding corn distillers dried grains with solubles on muscle quality traits and lipidomics profiling of finishing pigs”.

1.     Original submission.

1.1.  Recommendation

   Major revisions or resubmission

2.     Comments to the Authors

Manuscript Number: animals-2696708

Title: Effects of feeding corn distillers dried grains with solubles on muscle quality traits and lipidomics profiling of finishing pigs.

Overview and general recommendations:

The results may be interesting and may contribute to the knowledge of pork composition derived from alternative feedstuffs in the diet of the animals. This may be beneficial to plan nutritional strategies to produce nutritional pork for humans. The document is written with clarity and detail, as well as it is written in very good English. Besides, the authors have adequately employed very modern techniques to elucidate the chemical composition of lipids in meat such as lipodomics.

Despite the previous, I have two big concerns related to the design of the experiment. First, why the DDGS diet was allowed to be evidently superior in crude fat and crude fiber content? Due to this, the FAs profile in the meat cannot be certainly pointed if it was affected by the amount of fat to be deposited or by the differences in the fatty acid profile of the diet. Hence, the second observation has a high relevance. What was the FA composition of the diets? It is important to report it since one of the major aims of this study was to study the lipid composition of the diet. Why not analyze the diet to have definitive conclusions? Hence, I strongly suggest including the FAs analysis of the diets. Another variable of importance that should be measured to discuss the results is the dorsal fat thickness. Since the fat availability was higher in the DDGS diet, it was important to see if the animals were depositing differently the FAs in neutral and phospholipids.

I strongly suggest putting together the Results and Discussion. Unless the journal format strictly asks the opposite.

 Minor comments

Line 28: What do the authors mean by “These findings imply”?

Line 35. I suggest not to use apostrophes in scientific writing.

Line 84. What FAs profile are the authors reporting, the one in the total fat? Why not analyze it separately?

Line 86. What is “conventional” for the authors? Was the slaughter of the pigs regulated by any local, regional, or international law, conditions, etc?

Line 87. What was the size of the samples? Why “Some samples” and “other samples”? What were the criteria for doing this? A figure could help to explain this part.

Line 107. Rewrite the sentence.

Line 126. Avoid apostrophes. Rewrite the sentence.

Lines 126-135. It seems that authors are reporting the FAs profile of total fat. What FAs profile in meat is more susceptible to be modified by the animal diet, the phospholipids, or the neutral lipids? This is a major question to elucidate results.  

Line 133. “The iodine value was calculated…”

Lines 173-176. What is the reason for the b* increase in the pork of DDGS diets? Why not calculate C* and HUE?

Table 4. Why not report the individual concentrations of all the analyzed FAs? Why report only percentages? Why FAs such as; C20:4, C24:0, etc. are not listed individually? They need to be reported.

Line 216. TC is expected in adipocytes, nevertheless, the interesting would be to know what fractions are coming from membranes and what others are from neutral fat fractions.

Figure 4 should be presented in a way to see more clearly the text and values. The actual figure seems very small which makes it difficult to read.

Line 281. A statistical analysis that allows the integration of all the variables analyzed and describes possible interactions may be interesting, for both, elucidating the results and finding interactive ways among them.     

Comments on the Quality of English Language

I do not have comments. English has minor details to be corrected. 

Round 2

Reviewer 4 Report

Comments and Suggestions for Authors

Peer Review report 1 on: “Effects of feeding corn distillers dried grains with solubles on muscle quality traits and lipidomics profiling of finishing pigs”.

1.     Original submission.

1.1.  Recommendation

   Accepted

2.     Comments to the Authors

Manuscript Number: animals-2696708

Title: Effects of feeding corn distillers dried grains with solubles on muscle quality traits and lipidomics profiling of finishing pigs.

Overview and general recommendations:

The results are interesting and contribute to the knowledge of pork composition derived from alternative feedstuffs in the diet. This may be beneficial to plan nutritional strategies to produce nutritional and healthy pork for humans.

I recognize the big effort of the author to update and improve the whole document. The manuscript is now coherent and relevant. All my previous comments have been successfully attended. Hence, I consider that the manuscript is ready to be published.